# Effect of the Nanostructured Zn/Cu Electrocatalyst Morphology on the Electrochemical Reduction of CO_2_ to Value-Added Chemicals

**DOI:** 10.3390/nano11071671

**Published:** 2021-06-25

**Authors:** Piriya Pinthong, Phongsathon Klongklaew, Piyasan Praserthdam, Joongjai Panpranot

**Affiliations:** 1Center of Excellence on Catalysis and Catalytic Reaction Engineering, Department of Chemical Engineering, Faculty of Engineering, Chulalongkorn University, Bangkok 10330, Thailand; piriya.pin@hotmail.com (P.P.); pongsaton_putea@hotmail.com (P.K.); piyasan.p@chula.ac.th (P.P.); 2Bio-Circular-Green-economy Technology & Engineering Center, BCGeTEC, Department of Chemical Engineering, Faculty of Engineering, Chulalongkorn University, Bangkok 10330, Thailand

**Keywords:** Zn dendrite, bulky Zn, Zn/Cu electrode, electrodeposition, CO_2_ reduction, electrocatalysis

## Abstract

Zn/Cu electrocatalysts were synthesized by the electrodeposition method with various bath compositions and deposition times. X-ray diffraction results confirmed the presence of (101) and (002) lattice structures for all the deposited Zn nanoparticles. However, a bulky (hexagonal) structure with particle size in the range of 1–10 μm was obtained from a high-Zn-concentration bath, whereas a fern-like dendritic structure was produced using a low Zn concentration. A larger particle size of Zn dendrites could also be obtained when Cu^2+^ ions were added to the high-Zn-concentration bath. The catalysts were tested in the electrochemical reduction of CO_2_ (CO_2_RR) using an H-cell type reactor under ambient conditions. Despite the different sizes/shapes, the CO_2_RR products obtained on the nanostructured Zn catalysts depended largely on their morphologies. All the dendritic structures led to high CO production rates, while the bulky Zn structure produced formate as the major product, with limited amounts of gaseous CO and H_2_. The highest CO/H_2_ production rate ratio of 4.7 and a stable CO production rate of 3.55 μmol/min were obtained over the dendritic structure of the Zn/Cu–Na200 catalyst at −1.6 V vs. Ag/AgCl during 4 h CO_2_RR. The dissolution and re-deposition of Zn nanoparticles occurred but did not affect the activity and selectivity in the CO_2_RR of the electrodeposited Zn catalysts. The present results show the possibilities to enhance the activity and to control the selectivity of CO_2_RR products on nanostructured Zn catalysts.

## 1. Introduction

Removal of CO_2_, the principal greenhouse gas, from the atmosphere is critical in order to avoid climate disasters caused by global warming. While CO_2_ capture and storage face geological risks such as leakage and earthquakes, the sequestration and use of CO_2_ as a carbon feedstock for chemicals, fuels, and other derivative materials are an alternative feasible and economic pathway to recycle CO_2_ into various useful resources [1]. Among the different CO_2_ conversion routes, electrochemical reduction of CO_2_ (CO_2_RR) using electricity from renewable energies is one of the best currently available technologies that can meet the energy demand for CO_2_ reduction by low-carbon and low-cost electricity. A variety of carbon-containing products, such as formate, carbon monoxide, methane, and alcohol, can be derived from CO_2_RR in aqueous electrolytes [1,2,3,4]. Carbon monoxide (CO) is an interesting product of this process because it can be used as a reactant for the Fischer–Tropsch process to produce hydrocarbon fuels and chemicals [5,6]. Furthermore, it can be simply separated from liquid electrolytes.

There are various kinds of electrocatalysts that promote CO production in the electrochemical reduction of CO_2_, including both noble metals such as Au and Ag [7,8,9] and non-noble metals such as Cu, Sn, and Zn [10,11,12,13]. Noble metals are expensive, limiting their uses in large-scale commercial applications; thus, non-noble metals are preferred. Typically, a metal foil or plate is used as a cathode (working electrode) in the electrochemical reduction of CO_2_, but this exhibits low efficiency and selectivity toward CO production [14,15,16]. Using catalysts with a high surface area is a promising way to improve catalytic activity. Nanostructured materials have been employed in several works, such as nanocoral silver [17] and nanoporous ZnO [18], due to the large surface area, and a CO Faradaic efficiency above 90% can be achieved. For Zn-based catalysts, the performance of CO_2_RR to CO has been reported to be improved using nanostructured Zn electrocatalysts prepared by the electrodeposition method. The electrodeposited Zn catalysts, however, possessed different structures, such as a dendritic structure [19], a hexagonal structure [20], or a foam structure [21]. However, it is quite difficult to make a fair comparison among these studies to determine which morphologies of the Zn nanostructure would provide the best performance. A number of research works have also paid attention to the development of bimetallic catalysts, because higher performances can be obtained by modifying the structure and morphology of the catalysts [22,23,24]. Gue et al. reported that a AgZn bimetallic thin film can promote the CO_2_RR to CO with a Faradaic efficiency of 84.2% [24] due to the synergistic effect of Zn and Ag. In addition, the Zn structure was further modified by mixing Cu and Zn [21]. Zn/Cu alloy catalysts exhibit higher CO selectivity than pure Cu or pure Zn [25].

In this work, Zn/Cu electrocatalysts were prepared by electrodeposition of Zn on Cu foil (Zn/Cu) and electrodeposition of Zn and Cu on Cu foil (ZnCu/Cu) in an electrodeposition solution consisting of NaCl (Zn/Cu-Na, ZnCu/Cu-Na) or HCl (Zn/Cu-H, ZnCu/Cu-H) with different deposition times. These catalysts were evaluated in the electrochemical reduction of CO_2_ to higher-value chemicals. The morphology, surface composition, and crystalline structure of these catalysts were investigated by scanning electron microscopy–energy dispersive X-ray spectroscopy (SEM-EDX) and X-ray diffraction (XRD).

## 2. Experimental Section

### 2.1. Electrode Preparation (Electrodeposition of Zn/Cu Electrodes)

Cu and Zn electrodes were prepared by cutting commercial Cu (0.1 mm thick, 99.9999%) and Zn (0.1 mm thick, 99.994%) foil, from Alfa Aesar^®^ (Ward Hill, MA, USA), to a size of 10 × 25 mm^2^. The metal electrodes were physically polished with 800G sandpaper to remove the oxide surface and were then rinsed with deionized water before drying with nitrogen.

Nanostructured Zn/Cu and ZnCu/Cu electrodes with different morphologies were prepared by electrodeposition of ZnCl in either a NaCl or a HCl bath. A platinum rod (length 76 mm, diameter 2 mm) from Metrohm^®^ (Herisau, Switzerland) was used as the counter electrode (anode). A polished Cu electrode was used as a cathode in a solution containing 0.05 M ZnCl_2_ (Ajax Finechem Pty Ltd., New south whales, Australia) and 0.05 M NaCl (Sigma-Aldrich, St. Louis, MO, USA). Electric current was applied at 20 mA/cm^2^ for 60 and 200 s to obtain Zn/Cu–Na60 and Zn/Cu–Na200, respectively. Furthermore, the effect of copper ions on the electrodeposition was investigated by adding CuCl_2_ to the solution. A solution consisting of 0.05 M ZnCl_2_, 0.05 M NaCl, and 1.5 mM CuCl_2_ was employed under similar electrodeposition conditions to obtain ZnCu/Cu–Na60 and ZnCu/Cu–Na200.

Under acidic conditions (HCl bath), electrodeposition was performed in a solution consisting of 0.2 M ZnCl_2_ and 1.5 M HCl. Electric current was applied at 0.3 A/cm^2^ for 60 and 200 s to obtain Zn/Cu–H60 and Zn/Cu–H200, respectively. The effect of Cu ions was also studied using a solution consisting of 0.2 M ZnCl_2_, 1.5 M HCl, and 6 mM CuCl_2_ to obtain ZnCu/Cu–H60 and ZnCu/Cu–H200. The obtained electrodes were rinsed with deionized water and dried with nitrogen.

### 2.2. Characterization of Electrodes

The crystalline structure of electrocatalyst samples was analyzed using a SIEMENS D 5000 X-ray diffractometer (Munich, Germany) with a CuK_α_ radiation source (*λ* = 0.154439 nm) and nickel filtered in the 2θ degree range of 20°–80° (scan rate = 0.5 s/step). The morphology and surface composition of the electrocatalysts were analyzed by scanning electron microscopy–energy dispersive X-ray spectroscopy (SEM-EDX) using a Hitachi model S-3400N (Tokyo, Japan) scanning electron microscope.

### 2.3. Electrochemical Reduction of CO_2_

All experiments on the electrochemical reduction of CO_2_ were performed in an H-type cell at room temperature and atmospheric pressure, as shown in the schematic diagram in Figure 1. The cathodic and anodic parts were separated by a Nafion^®^ 117 membrane (Sigma-Aldrich, St. Louis, MO, USA). The experiments were carried out in a three-electrode cell system. A silver/silver chloride (Ag/AgCl) electrode from Metrohm^®^ (Herisau, Switzerland) was used as the reference electrode. The counter electrode was a platinum foil (0.1 mm thick, 99.997%) from Alfa Aesar (Ward Hill, MA, USA). The working electrodes (Zn foil, Cu foil, and the electrodeposited electrodes) were immersed in an electrolyte solution with a geometrical area of 1 cm^2^. The catholyte and anolyte were 20 mL of 0.1 M KHCO_3_ solution. The electrolyte was saturated with 100 mL/min of CO_2_ gas for 30 min with a CO_2_ flow rate of 20 mL/min during the reaction. The reduction potential was controlled by a potentiostat during a reaction time of 70 min. The gaseous products were analyzed by an online gas chromatography system with a thermal conductivity detector (TCD). The liquid-phase products were identified and quantified using the NMR technique. The electrocatalysts with outstanding performances were further investigated at different reduction potentials from −1.4 to −2.0 V vs. Ag/AgCl, and a stability test was performed at the appropriate potential for 4 h.

## 3. Results and Discussion

### 3.1. Electrocatalyst Characterization

The Zn/Cu and ZnCu/Cu electrocatalysts in this study were prepared by electrodeposition techniques under different conditions in order to obtain different morphologies of the deposited metal. The method allows successful deposition of 2D and 3D metal nanostructures on the substrates. Grain sizes of the deposited metals in the nanometer range were obtained by selecting electrodeposition variables (e.g., bath composition, pH, temperature, current density) such that nucleation of new grains was favored rather than growth of existing grains [26]. SEM images of the electrocatalyst samples are shown in Figure 2. A rock-like surface with some streaks was observed on the mechanically polished Zn and Cu foil (Figure 2a,b). After electrodeposition, a fern-like dendritic structure of Zn nanoparticles was obtained on Zn/Cu–Na60 and Zn/Cu–Na200, as shown in Figure 2c,d, respectively. The high deposition rate of Zn for 0.05 M ZnCl_2_ and the lower-concentration region of Zn ions at the electrocatalyst surface forced the dendritic structures to grow outward toward the higher-concentration region of Zn ions [19]. The deposition time (60 and 200 s) did not affect the morphology of Zn/Cu–Na, as the morphologies of Zn/Cu–Na60 and Zn/Cu–Na200 were similar. When Cu ions were added, the dendritic structures were not much altered, as seen in Figure 2e for ZnCu/Cu–Na60 and Figure 2f for ZnCu/Cu–Na200, probably due to the low concentration of CuCl_2_ (0.0015 M) in the deposition bath. In contrast, bulky (hexagonal) structures with average particle sizes of 1–4 and 4–10 μm were obtained on Zn/Cu–H60 and Zn/Cu–H200, respectively (Figure 2g,h). It is suggested that a bulky structure is formed because of the high ZnCl_2_ concentration (0.2 M) used in the deposition bath [19]. Increasing the deposition time resulted in an increased average particle size of Zn/Cu–H. Moreover, when Cu ions were simultaneously added to the HCl bath with Zn ions, another form of dendritic structure was observed, as shown in Figure 2i,j for ZnCu/Cu–H60 and ZnCu/Cu–H200, respectively. Although the deposition time did not have much impact on the dendritic morphology of ZnCu/Cu–H, adding Cu ions to the solution changed the morphology from a bulky structure to a dendritic structure as the concentration of Zn ions near the electrocatalyst surface was diluted by the presence of Cu ions. Cu ions can compete with Zn ions for deposition on a Cu foil. The deposited Cu can also accelerate the formation of hydrogen gas during the electrodeposition process. In addition, Cu has a higher hydrogen bond strength than Zn [21]. The hydrogen gas that is produced during the electrodeposition process also competes with Zn ions. It is also noticed that the particle size of the dendritic structure of ZnCu/Cu–H is bigger than that of ZnCu/Cu–Na due to the higher concentration of ZnCl_2_ in the deposition bath.

The surface compositions of the deposited electrodes are shown in Table 1. At a low Zn ion concentration in the NaCl bath, although increasing the deposition time did not affect the morphology of the fern-like dendritic structure being formed, the percentages of Zn increased from 75% to 93% as the deposition time increased from 60 to 200 s. Adding Cu did not have much impact on the amount of Zn being deposited under the conditions used (60 s, NaCl bath). Conversely, at a high Zn ion concentration in the HCl bath, there was little effect of the deposition time, whereas a lower amount of Zn was obtained when ZnCl_2_ and CuCl_2_ were simultaneously deposited.

The XRD results of all the electrocatalysts are shown in Figure 3. The XRD patterns of the deposited electrocatalysts matched perfectly with metallic Zn and Cu according to the JCPDS database (Cu: JCPDS 04-0836; Zn: JCPDS 00-004-0831). Except for the Zn foil, a strong characteristic peak corresponding to the Cu(200) facet was apparent at a 2θ degree of 50.1° [25]. The intensity of the Cu(200) peak decreased after deposition of Zn nanostructures on the Cu foil. The diffraction peaks at 36.3° and 43.3° (2θ) confirmed that Zn with a (101) and a (002) lattice structure was properly deposited on the surface of the Cu foil. The crystal planes (100) and (102) of Zn became more visible for Zn/Cu-–H200, in which the highest amount of Zn was deposited on the substrate. For the higher Zn concentration in the HCl bath, additional peaks corresponding to the Cu_4_Zn phase were detected on both ZnCu/Cu–H60 and ZnCu/Cu–H200 [21], indicating the formation of CuZn alloy by simultaneous deposition of Cu and Zn ions. Such result was not observed under low-Zn-concentration conditions (i.e., ZnCu/Cu–Na60 and ZnCu/Cu–Na200), and the addition of Cu^2+^ during electrodeposition neither changed the morphologies nor changed the structure of Zn being deposited.

### 3.2. Electrocatalytic Performances in CO_2_RR

All the deposited electrodes were tested in the electrochemical reduction of CO_2_ at a potential of −1.6 V vs. Ag/AgCl for 70 min. The production rates for both gaseous and liquid products and CO Faradaic efficiency are shown in Table 2 and Figure 4, respectively. Both Zn foil and Cu foil can produce CO, formate, and n-propanol as the products from CO_2_RR, while H_2_ gas was produced by the hydrogen evolution reaction. Typically, a Cu foil exhibits higher activity than a Zn foil because its resistance is lower, resulting in a faster electron transfer rate [27]. Cu also has a lower theoretical limiting potential for formate production than Zn [28]. Furthermore, the higher binding energy of the Cu–CO bond leads to a higher production rate of n-propanol on a Cu foil than a Zn foil [29].

All the electrodeposited Zn catalysts with a dendritic structure (prepared under both low and high Zn ion concentrations) exhibited much higher CO production rates (~3–4.7 μmole/min) compared to the Zn foil (0.9 μmole/min) due to the higher surface area. Among them, the CO production rate increased with increasing Zn deposited. The CO/H_2_ production rate ratio ranged between 2.5 and 4.7, which was not much different from that of the Zn foil (4.1). It has been suggested that the dendritic structure contains a higher density of stepped sites that can suppress hydrogen evolution [19]. Among the various electrocatalysts, the highest CO/H_2_ production rate of 4.7 and a CO Faradaic efficiency of 83.2% were obtained on Zn/Cu–Na200 as it maintained the fern-like dendrite structure with a high amount of Zn. Furthermore, the CO Faradaic efficiency compared with previous works is shown in Table 3. The CO Faradaic efficiency of the dendritic Zn structure in this work was comparable to those reported in the literature and was relatively high comparing to the non-dendritic structure in a bicarbonate electrolyte. The results emphasize the effect of Zn morphology on the activity of Zn electrocatalysts. The CO:H_2_ ratio was relatively high compared to other Zn dendrite electrodes, as reported in the literature [19,30]. The production rates of formate (0.07–0.16 μmole/min) and n-propanol (0.02–0.06 μmole/min) over the Zn foil and the Zn dendrite electrocatalysts were essentially similar, despite the different conditions of the electrodeposition bath used. It should also be noted that the low coverage of Zn particles (as found in Zn/Cu–Na60 and ZnCu/Cu–Na60) and the presence of Cu_4_Zn alloy (as found in ZnCu/Cu–H60 and ZnCu/Cu–H200) did not have much influence on the selectivities of CO_2_RR products of the Zn-based electrocatalysts. Cu_4_Zn alloy can promote the formation of ethylene and ethanol, as reported by Ren et al. [27]. However, it was probably not the dominant phase in this study. Although the formation of other liquid hydrocarbon products such as methane and methanol has been reported on dendritic Zn [30], a Ag foam substrate with some certain crystalline structures and much higher over-potential is necessary in order to strongly bind CO* on the surface.

Unlike the dendritic structure Zn, the production rate of formate was drastically enhanced by more than five-fold on the electrodeposited Zn catalysts with bulky morphology, such as Zn/Cu–H60 and Zn/Cu–H200, compared to the Zn foil. However, there was little effect of the particle size of bulky Zn electrocatalysts (in the range of 1–10 μm) on the CO_2_RR activity and selectivity of the products. It is likely that the hexagonal close-pack structure of bulky Zn promotes formate production. According to a DFT study on various metals by Yoo et al., both COOH* (carboxyl) and HCOO* are presented as the intermediates of CO_2_RR, but the selective reduction of CO_2_ to formate (HCOOH) is more likely to occur via the HCOO* intermediate than the COOH* intermediate, which further reduces to CO and H_2_ [28]. The present results suggest that the pathway through HCOO* to formate is dominant on a bulky zinc structure, as depicted in the proposed scheme in Figure 5. Recently, cation vacancies (V_Zn_) in the ZnS structure have been proposed to be new active sites on the ZnS surface that can stabilize HCOO* via O bridging between the intermediate and zinc ion vacancies V_Zn_ and hence is more efficient for the production of formate [31]. From these findings, it is speculated that the bulky zinc structure contains a high amount of cation vacancies, leading to the high production rate of formate, with limited formation of CO and H_2_. The absence of n-propanol on bulky Zn could be related to the high coverage of Zn species on the surface. As revealed by the EDX results, Zn/Cu–H60 and Zn/Cu–H200 showed the lowest percentages of Cu on the substrate (2–4 wt%). Typically, the formation of n-propanol occurs via C–C coupling of adsorbed-CO and adsorbed-methane intermediates on the Cu surface [32].

To further investigate the electrochemical behavior of the Zn/Cu–Na200 catalyst in CO_2_RR, linear sweep voltammetry (LSV) was performed. According to the LSV results in Figure 6, the Zn/Cu–Na200 catalyst was tested in CO_2_RR at different over-potentials of −1.4, −1.8, and −2.0 V vs. Ag/AgCl. The activity and selectivity of the Zn/Cu–Na200 catalyst are given in Table 4. At −1.4 V vs. Ag/AgCl, the CO formation rate was fairly low and surprisingly formate was not detected. Thermodynamically, the electron requirement for the half-cell reaction for n-propanol formation is 18 electrons, much higher than CO and formate production, which requires only 2 electrons [33]. The present results suggest that the production of formate may occur via a different pathway than the formation of CO and n-propanol on the Zn electrocatalysts. Increasing the over-potential from −1.4 to −1.6 resulted in a higher CO/H_2_ production rate, whereas a further increase in the potential beyond −1.6 V vs. Ag/AgCl led to lower CO/H_2_ production. Generally, the electron transfer rate at higher potential is faster, whereas the CO_2_ transfer rate remains unchanged. It has been suggested that this mass transfer limitation causes H_2_ formation [34]. The production rates of gaseous products monotonically increased as the over-potential increased from −1.4 to −2.0 V vs. Ag/AgCl, with only a slight increase in liquid products.

The stability test results of Zn/Cu–Na200 are shown in Figure 7. The formation rate of CO was comparatively stable around 3.55 μmol/min throughout the 4 h reaction time. The morphologies of the electrocatalysts before and after the stability test are shown in Figure 8. It is clearly evident that the particle size of the Zn dendrite increased from 0.2–0.5 to 0.5–1.2 μm, as shown in Figure 8a,b, respectively, indicating that dissolution and re-deposition of Zn nanoparticles occurs during CO_2_RR [19]. Nevertheless, the percentage of Zn remained the same at around 93 wt%. In addition, it was confirmed that the particle size or shape of the dendritic structure of Zn did not affect the activity and selectivity of the CO_2_RR of the electrodeposited Zn catalysts.

## 4. Conclusions

In this study, Zn/Cu and ZnCu/Cu electrocatalysts were prepared via the electrodeposition method with various bath compositions and deposition times. A low Zn concentration (e.g., 0.05 M ZnCl_2_ in NaCl solution or 0.2 M ZnCl_2_ in HCl solution in the presence of Cu^2+^) enabled mass transfer limited the growth of well-defined Zn dendrite structures. All the Zn dendrites exhibited higher activities in CO_2_RR than the Zn foil. The highest CO/H_2_ production rate (4.7) was obtained over Zn/Cu–Na200 having a fern-like dendritic structure and a relatively high amount of Zn being deposited. However, at the same over-potential, the activities and product selectivities of the Zn dendrites were essentially similar, regardless of the particle size/shape of the dendritic structure. On the contrary, the bulky Zn structure with an average particle size of 1–10 μm obtained from the high-Zn-concentration bath strongly favored the production of formate with limited amounts of gaseous CO and H_2_ products. The dissolution and re-deposition of Zn nanoparticles during the 4 h of reaction led to a larger particle size of the Zn dendritic structure but did not affect the activity and selectivity of the CO_2_RR of the electrodeposited Zn catalysts.

## Figures and Tables

**Figure 1 nanomaterials-11-01671-f001:**
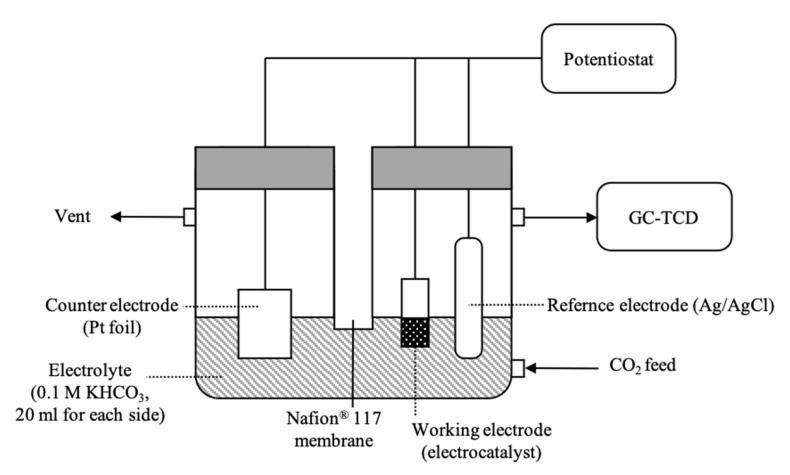
Schematic electrochemical CO_2_ reduction setup.

**Figure 2 nanomaterials-11-01671-f002:**
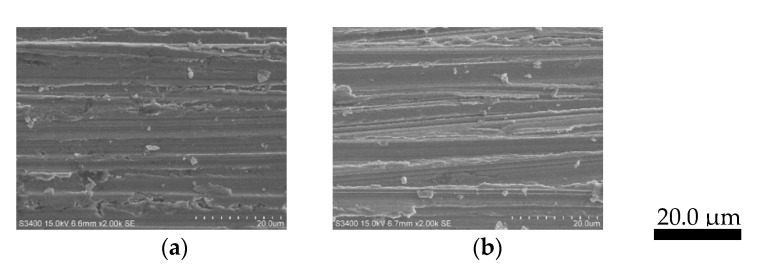
SEM images of (**a**) Zn foil, (**b**) Cu foil, (**c)** Zn/Cu–Na60, (**d**) Zn/Cu–Na200, (**e**) ZnCu/Cu–Na60, (**f**) ZnCu/Cu–Na200, (**g**) Zn/Cu–H60, (**h**) Zn/Cu–H200, (**i**) ZnCu/Cu–H60, and (**j**) ZnCu/Cu–H200.

**Figure 3 nanomaterials-11-01671-f003:**
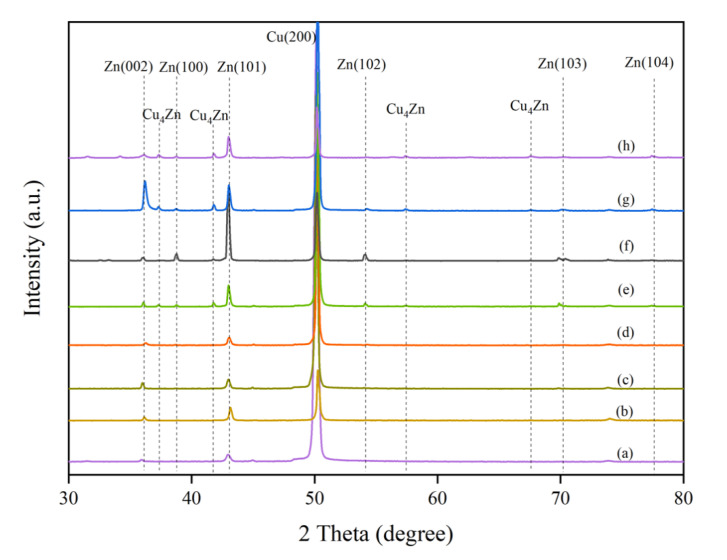
XRD patterns of (a) Zn/Cu–Na60, (b) Zn/Cu–Na200, (c) ZnCu/Cu–Na60, (d) ZnCu/Cu–Na200, (e) Zn/Cu–H60, (f) Zn/Cu–H200, (g) ZnCu/Cu–H60, and (h) ZnCu/Cu–H200.

**Figure 4 nanomaterials-11-01671-f004:**
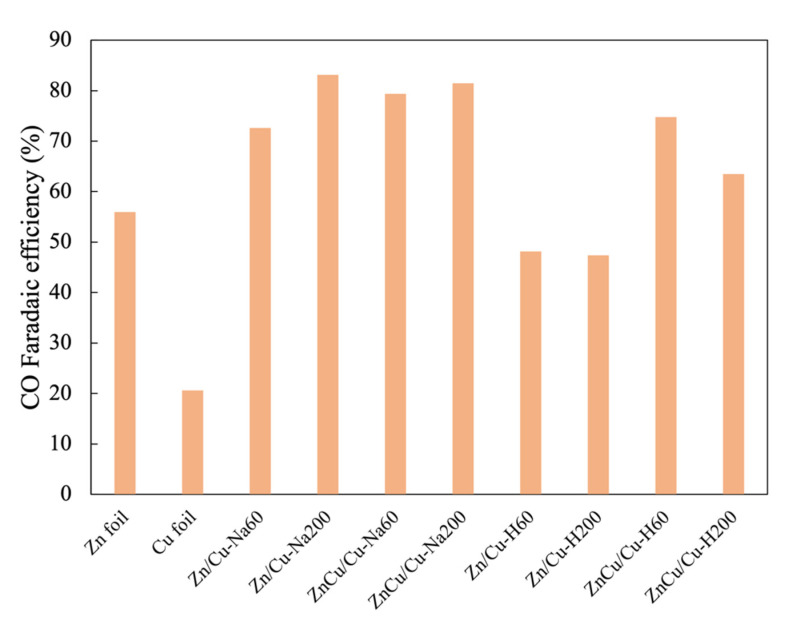
CO Faradaic efficiency measure at −1.6 V vs. Ag/AgCl.

**Figure 5 nanomaterials-11-01671-f005:**
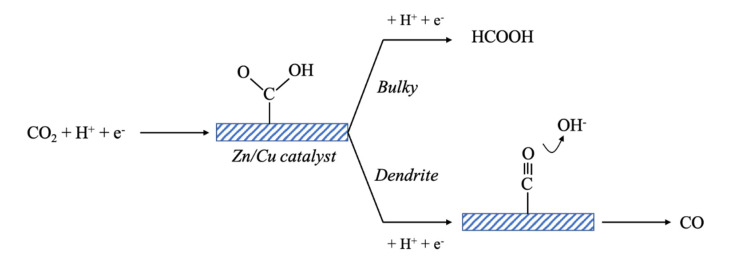
Proposed reaction pathways for CO_2_RR over dendritic and bulky structures of Zn/Cu catalysts.

**Figure 6 nanomaterials-11-01671-f006:**
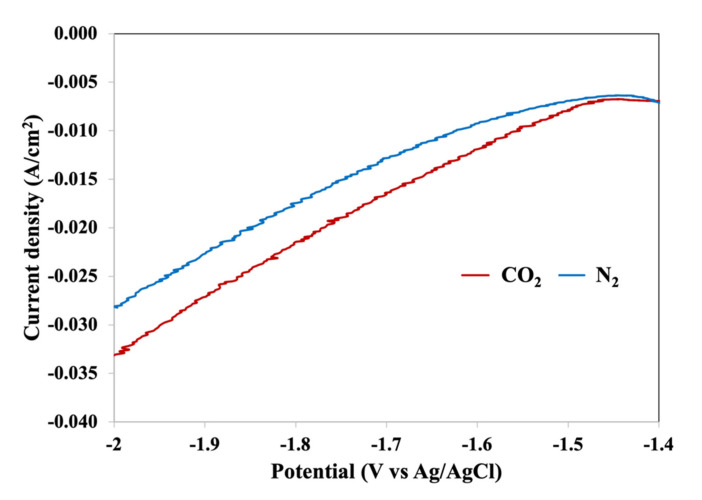
LSV results of the Zn/Cu–Na200 catalyst.

**Figure 7 nanomaterials-11-01671-f007:**
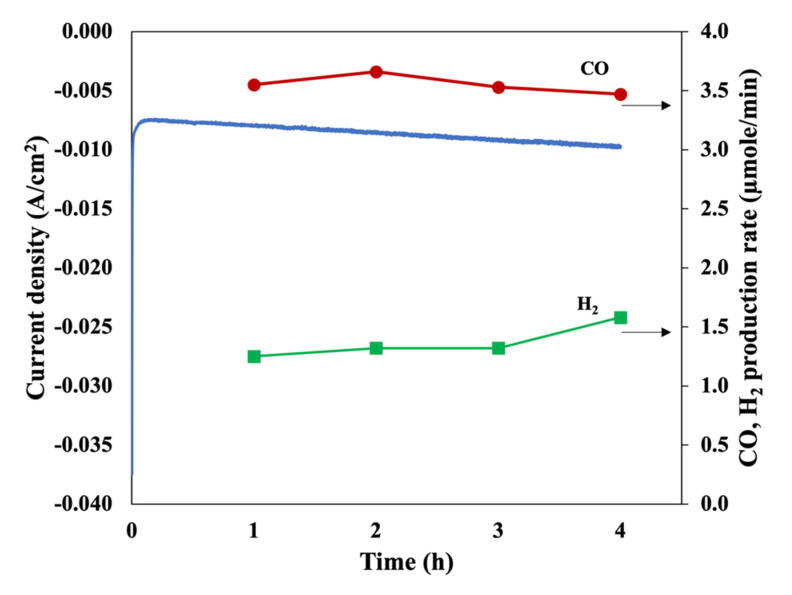
Catalytic performance of Zn/Cu–Na200 at −1.6 V vs. Ag/AgCl for 4 h.

**Figure 8 nanomaterials-11-01671-f008:**
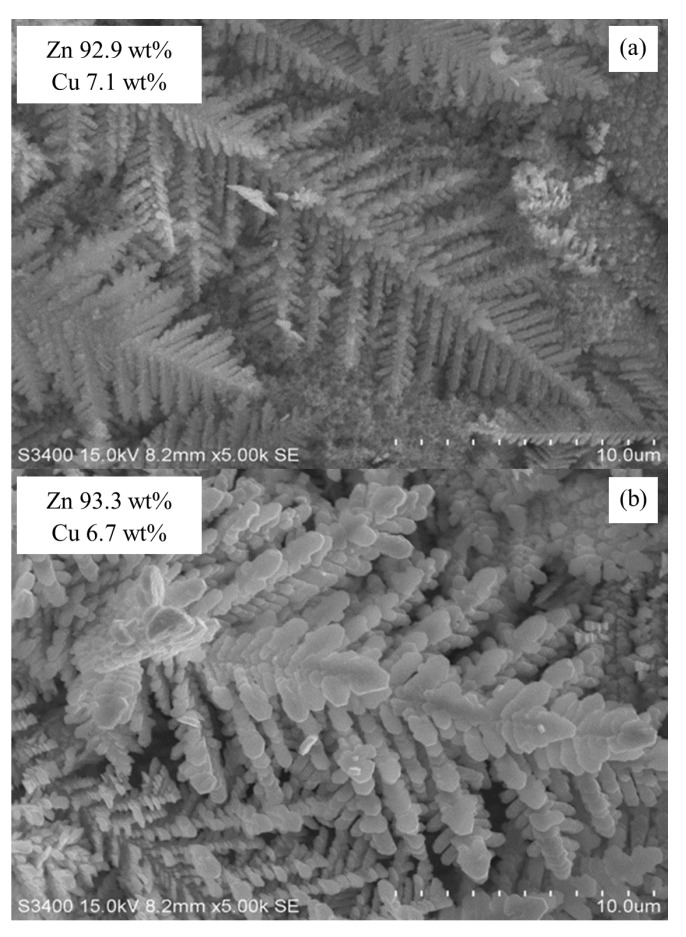
SEM images of Zn/Cu–Na200 (**a**) before and (**b**) after the stability test.

**Table 1 nanomaterials-11-01671-t001:** Percentage by weight of deposited metals on the electrocatalysts.

Electrocatalyst	Percentage by Weight	
	Zn (%)	Cu (%)
Zn/Cu–Na60	74.9	25.1
Zn/Cu–Na200	92.9	7.1
ZnCu/Cu–Na60	72.5	27.5
ZnCu/Cu–Na200	93.1	6.9
Zn/Cu–H60	96.0	4.0
Zn/Cu–H200	98.1	1.9
ZnCu/Cu–H60	86.1	13.9
ZnCu/Cu–H200	88.8	11.2

**Table 2 nanomaterials-11-01671-t002:** Catalytic performances of electrocatalysts with different deposition times at a reaction potential of −1.6 V vs. Ag/AgCl for 70 min.

Electrocatalyst	Production Rate (μmol/min)				CO/H_2_ Production Rate Ratio	Average Electrochemical Charge Passing per Minute (C/min)
	CO	H_2_	Formate	n-Propanol		
Zn foil	0.87	0.21	0.07	0.04	4.1	0.15
Cu foil	0.96	2.21	0.48	0.15	0.4	0.45
Zn/Cu–Na60	2.93	0.92	0.13	0.06	3.2	0.39
Zn/Cu–Na200	3.61	0.77	0.09	0.06	4.7	0.42
ZnCu/Cu–Na60	3.28	0.93	0.05	0.04	3.5	0.40
ZnCu/Cu–Na200	4.72	1.29	0.06	0.03	3.7	0.56
Zn/Cu–H60	0.65	0.45	0.36	-	1.5	0.13
Zn/Cu–H200	1.01	0.54	0.36	-	1.8	0.21
ZnCu/Cu–H60	3.21	0.89	0.13	0.02	3.6	0.41
ZnCu/Cu–H200	4.14	1.63	0.16	0.02	2.5	0.63

**Table 3 nanomaterials-11-01671-t003:** Comparison of CO Faradaic efficiency on different Zn electrocatalysts in CO_2_RR.

Zn Electrocatalyst	Electrolyte	Potential (vs. Ag/AgCl)	CO Faradaic Efficiency (%)	Ref.
Zn/Cu–Na200	0.1 M KHCO_3_	−1.60	83.2	This work
ZnCu/Cu–Na200	0.1 M KHCO_3_	−1.60	81.5	This work
Zn dendrite	0.5 M NaHCO_3_	−1.72	79	[33]
Reduced nanoporous ZnO	0.25 M K_2_SO_4_	−1.66	92	[18]
Dendrite PD–Zn/Ag foam	0.1 M KHCO_3_	−1.80	76.4	[30]
Nano–Zn	0.5 M NaHCO_3_	−1.47	57	[14]

**Table 4 nanomaterials-11-01671-t004:** Performance of Zn/Cu–Na200 in CO_2_RR at various potentials for 70 min.

Entry	Potential (V) vs. Ag/AgCl	Production Rate (μmol/min)				CO/H_2_ Production Rate Ratio
		CO	H_2_	Formate	n-Propanol	
1	−1.4	1.08	0.48	-	0.02	2.3
2	−1.6	3.61	0.77	0.09	0.06	4.7
3	−1.8	6.98	2.95	0.10	0.06	2.4
4	−2.0	7.97	10.18	0.19	0.06	0.8

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
