# Peer review of "Effect of the Nanostructured Zn/Cu Electrocatalyst Morphology on the Electrochemical Reduction of CO2 to Value-Added Chemicals"

_nanomaterials, 2021, doi:10.3390/nano11071671_

Round 1
Reviewer 1 Report
The manuscript need major revisions in order to be published into the Journal of Nanomaterials. Please find attached my comments.

Reviewer 2 Report
In this manuscript, the authors electrodeposited various Zn/Cu nanostructures having different morphologies and they probed the products issued from the CO2 reduction reaction (CO2RR). Particularly they found out that the CO2RR products formed depend mainly on the electrocatalyst structure. In my opinion, this work is interesting and should be published in nanomaterials after the considering the following comments/suggestions.
-In ref 25, it was probed the formation of methanol and methane as products issued from the CO2RR at Zn dendritic materials whereas herein the production of such products were not reported. Could the authors comment on this?
-The cumulating production rates of the products reported in table 2 are found different from one electrocatalyst to another one. For instance using Zn foil, the cumulating production rate of the products (CO + H2 + formate + Propanol) is 1.19 µmol/min whereas using ZnCu/Cu-Na200 it is 6.1 µmol/min. The authors attributed this difference to the surface area change of the electrocatalysts. A higher surface area should lead to an increase of the electrocatalytic current. Therefore, I strongly encourage the authors to compare their production rates to the electrochemical current response. Particularly the electrochemical charge passing per minute for each electrocatalyst should be provided in table 2.
-The restructuration of the Zn dendrite observed is an interesting point raised by the authors. I think they should emphasize this observation on the abstract and conclusions. Is this effect also visible on the other nanostructures such as on ZnCu-H200?
Reviewer 3 Report
In this paper the authors present an effort to check THE Effect of the nanostructured Zn/Cu electrocatalyst morphology on the electrochemical reduction of CO2 to value-added chemicals. The work is nice and should be published.Author Response
Please see the attachment.

Round 2
Reviewer 1 Report
In the current form the manuscript is acceptable for publication
